

# Biology, ecology, and biogeography of eremic praying mantis *Blepharopsis mendica* (Insecta: Mantodea)

Zohreh Mirzaee[1,2,3], Marianna V.P. Simões[4], Roberto Battiston[5], Saber Sadeghi[2], Martin Wiemers[1] and Thomas Schmitt[1,3]

[1] Senckenberg German Entomological Institute, Müncheberg, Germany
[2] Biology Department, Faculty of Sciences, Shiraz University, Shiraz, Iran
[3] Entomology and Biogeography, Institute of Biochemistry and Biology, Faculty of Science, University of Potsdam, Potsdam, Germany
[4] Senckenberg Research Institute and Natural History Museum, Frankfurt am Main, Germany
[5] Museo di Archeologia e Scienze Naturali "G. Zannato", Montecchio Maggiore, Italy

Corresponding author
Zohreh Mirzaee,
zmirzaee1988@gmail.com

## ABSTRACT

**Background**. *Blepharopsis mendica* (Fabricius, 1775) is a large mantid species found from the Canary Islands across North Africa, the Middle East, and Pakistan. Research on this species has been limited, especially in Iran, despite the country's potential significance for studying its biology and distribution. Adults of this species are easily recognizable by their marble-white pattern and rhomboidal leaf-like pronotum. They are sit-and-wait predators that inhabit various open environments.

**Methods**. Field observations were conducted across various regions of the Egyptian Flower mantis (*Blepharopsis mendica*) global distribution, with a focus on Morocco, Tunisia, and Iran. Distribution data for *B. mendica* were gathered from fieldwork, museum collections, online biodiversity databases, and publications, totaling 593 occurrence points. Ecological niche modeling was performed using environmental data, and various models were evaluated for suitability. Phylogeographic analyses involved DNA sequencing and construction of a haplotype network to examine genetic relationships between populations. Divergence time estimation and biogeographical range expansion models were applied to explore historical distribution shifts of the species across different regions. The study provided comprehensive insights into the biology, distribution, and genetic history of *B. mendica*.

**Results**. We provide information on the life cycle, ootheca, defense behavior, habitat, and biogeography of the Egyptian Flower mantis *Blepharopsis mendica*. This mantid is an overwintering univoltine species with nymphs emerging in summer and becoming adults in spring. In the wild, females start oviposition in April and can lay their first ootheca within a week after mating. The species is distributed from the Canary Islands to Pakistan in the dry belt. Thus, its distribution is associated with xeric areas or desert and semi-desert habitats. Phylogeographic analyses revealed three major genetic lineages, (i) in the Maghreb, (ii) from Egypt via Arabia to Iran (with internal substructures), and (iii) likely in Pakistan; the estimated onset of differentiation into these lineages is of Pleistocene age. Defense behavior involves flying away or extending wings broadly and lifting forelegs. Performing laboratory breeding, we documented life cycle and color changes from first instar to adulthood. Due to overwintering, the last larval instar needs considerably longer than the others. At 25 °C ($\pm 2$), average adult life span was 118 days

(±6 SD) for females (range: 100–124) and 46 days (±5 SD) for males (range: 39–55), with a significant difference among sexes. On average, oothecae contained 32.3 eggs (±10.1 SD) and the mean incubation period was 36.8 days (±2.9 SD). We did not find evidence of parthenogenesis. In general, the biology of *B. mendica* shows a variety of adaptations to its often extreme and little predictable type of habitat.

## INTRODUCTION

Praying mantids occupy an important ecological niche, playing vital roles as predators. These creatures are renowned for their distinctive appearance and predatory prowess, wielding their forelegs with precision to capture and subdue a wide array of prey, including other insects, small fauna, and even their own kind. In this intricate web of life, their presence underscores the delicate balance and the indispensable role of these fierce predators in maintaining the equilibrium of insect populations within many of the world's diverse ecosystems (*Sampaio et al., 2009*).

The color changes of different stages of mantids also provide insights into their ecology and behavior. For example, coloration may play a role in camouflage, mate selection, or predator avoidance, and understanding these factors can help us to better understand the role that these species play in their ecosystems (*Battiston & Fontana, 2010*; *O'Hanlon, Li & Norma-Rashid, 2013*; *Wang, Wu & Zhao, 2017*).

One rather spectacular mantid species is the Egyptian Flower mantis *Blepharopsis mendica* (Fabricius, 1775). This large species is found from the Canary Islands throughout North Africa and the Middle East to Pakistan (*Battiston & Fontana, 2010*). Adults can be distinguished by their marble-white pattern all over their bodies and the rhomboidal leaf-like shape of their pronotum. This mantid, a sit-and-wait predator, inhabits open areas where it lives in green and dried shrubs. It exhibits exceptional camouflage with its cryptic shape, color, and behavior (Fig. 1), making it difficult to spot in its natural habitat (*Battiston & Fontana, 2010*).

Although *B. mendica* is a fascinating mantid, only two relatively old studies (*i.e.,* (*Korsakoff, 1934*; *Korsakoff, 1935*) dealt in more detail with the species' life cycle, biology, and other ecological aspects, while the more recent publications mostly address its distribution, taxonomic treatment by synonymizing *B. mendica nuda* Giglio-Tos, 1917 with the nominate subspecies, or are only presenting new faunistic records (*Ehrmann, 2011*; *Caesar et al., 2015*; *Panhwar et al., 2020*; *Nasser et al., 2021*). In particular, for Iran, there are practically no studies concerning the biology and distribution of *B. mendica*, and only records of this species from some parts of the country (Lorestan and Fars provinces) have been published so far (*Mirzaee & Sadeghi, 2019*; *Mirzaee & Sadeghi, 2021*). However, Iran, with its strikingly diverse array of ecosystems and hence high diversity of (often endemic) insect species (*Zehzad, Kiabi & Madjnoonian, 2002*), is a particularly important region for the study of *B. mendica*, primarily owing to its geographically extended arid and

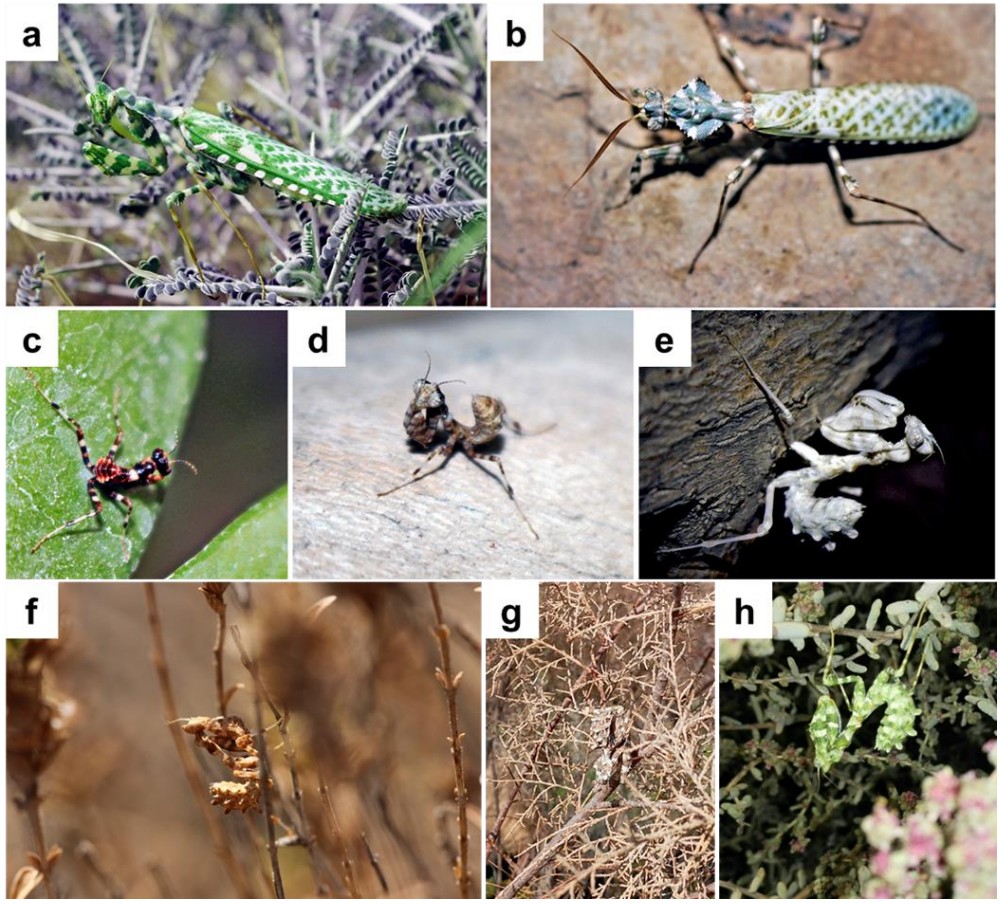

**Figure 1** *Blepharopsis mendica*: (A) female and, (B) male adult habitus, as well as (C) first, (D) second, and (E) forth instar larvae, (F, G, H) different instars larvae showing colour differences and the camouflage in their natural habitat.

semi-arid landscapes, often characterized by scrub vegetation, the ecosystems where this species typically thrives.

Here, we provide detailed information regarding the color change of *B. mendica* along its development under optimal laboratory conditions, its biology, life cycle, and behavior. Defensive behaviors of individuals in the wild are also documented and discussed. New data on the distribution of the species across Iran, where the species still is rather poorly studied, is presented together with additional information on its life history in the wild. These data in conclusion allow a more comprehensive understanding of the species' biology including its life history, ecology, evolution, distribution, and historical biogeography.

# MATERIALS AND METHODS

## Collecting and observation in the wild

Field observations have been done along the global distribution of this species to contextualize the data in a wider perspective. Wild specimens were observed and
documented in three focal points of the global distribution of this species: western habitats in Morocco, central habitats in Tunisia, and eastern habitats in Iran. Individuals of *B. mendica* from nine regions in five different provinces of Iran (Lamerd, Fasa, Shiraz, Fars province; Jam, Soroo, Tombak, Busheher province; Khomeini Shahr, Isfahan province; Abadan, Khozestan province; Eshkanan, Hormozgan province) were collected during field surveys from 2019 to 2021. The presence of individuals and their defense behavior were observed and photographed within natural habitats. Three oothecae of this species were collected from branches of trees or bushes in Darab, Fars Province, and Jam, Bushehr Province, during June and July 2020, but they were empty and already hatched at the time of collecting. Species and ootheca identification were carried out following *Battiston & Fontana (2010)*. All materials collected during this survey are preserved in the following collections: Zohreh Mirzaee private collection, Müncheberg, Germany (ZMPC); Zoological Museum of Shiraz University, Shiraz, Iran (ZM-CBSU); and Mantodea collection of Senckenberg German Entomological Institute, Müncheberg, Germany (SDEI).

## Rearing and laboratory condition

From two adult individuals collected from xeric shrublands of Bushehr province (27°50′37.35″N, 52°03′51.92″E), one female laid one ootheca, which was kept in a glass jar (15 × 15 × 10 cm) at room temperature (25–27 °C). The relative air humidity (RH) was maintained at 40–45% with water misted on a regular basis. A digital terrarium hygrometer (HTC2) (Dongguan City, China) was used to measure RH.

The hatched nymphs were kept in separate glass jars (6 × 6 × 4 cm) during the first and second instar, thereafter transferred to bigger jars (12 × 12 × 10 cm). The jars containing the nymphs were maintained at 33–35 °C. One stick was placed in each jar to assist molting. Ventilation was enabled by three holes (two mm in diameter each) in the lid of the jars. During the first and second instar, nymphs were fed with fruit flies (*Drosophila melanogaster* Meigen, 1830), two to three individuals per nymph every three days. Later instars were fed with living mealworm larvae (*Tenebrio molitor* Linnaeus, 1758), small living grasshoppers (*Calliptamus* spec.), moths (mostly *Eupithecia* spec. and *Leucania* spec.) and house flies (*Musca domestica* Linnaeus, 1758) twice a week.

All jars were checked daily. We recorded all information regarding the dates of molting and number of molts. To prevent contamination or disturbance, we removed all unfinished or dead prey. The sex of each individual was noted after the last molt. The adults were used for further rearing, testing different conditions (*i.e.,* mated, not mated).

## Data analyses of rearing

We calculated the mean number of days (with their standard deviations) between molts and adulthood (based on nymphs reaching the adult phase), separately for males and females. To assess the difference in mean adult longevity between males and females, a two-sample *t*-test was conducted. The *t*-test is appropriate for comparing the means of two independent groups. Statistical analysis was performed using RStudio 3.6.3 (*RStudio Team, 2021*; *R Core Team, 2021*) with the base R package. Oothecae resulting from the first generation bred in captivity were measured, and the numbers of egg chambers inside

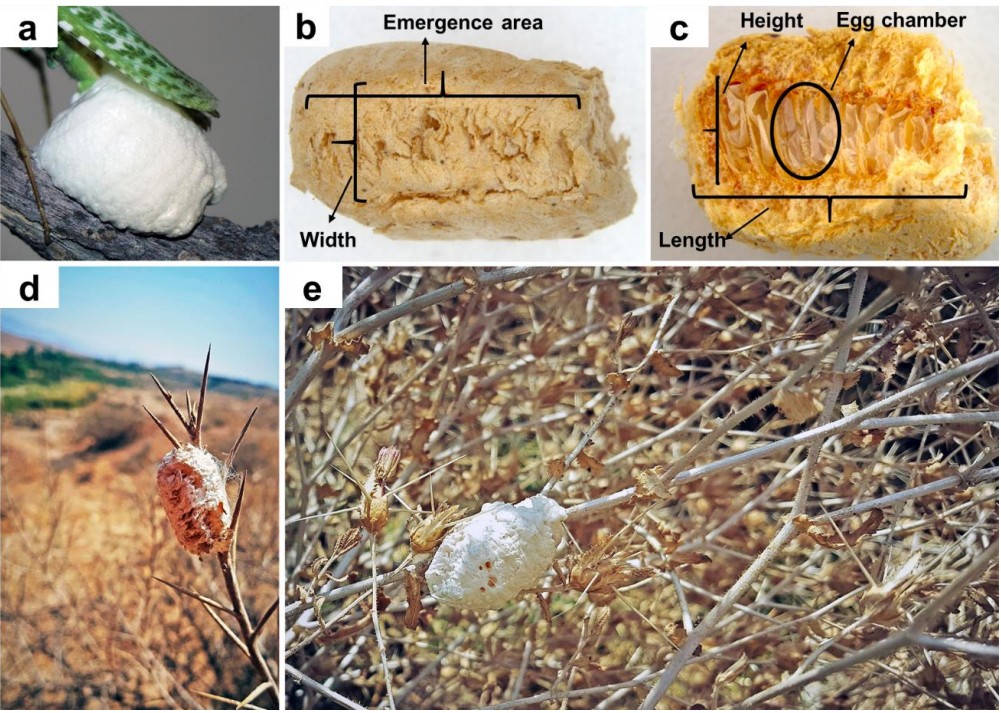

**Figure 2** *Blepharopsis mendica* oothecae: (A) fresh ootheca, (B) dorsal view, (C) dissected ootheca, and (D, E) in natural habitat.

fertilized and unfertilized oothecae were counted. Based on descriptions provided by *Brannoch et al. (2017)*, the length, width, and height of each ootheca were assessed. To count the number of eggs per ootheca, they were dissected along their length and examined under a LEICA M205 C binocular microscope. The ootheca parameters were measured as shown in Figs. 2A and 2B. A digital camera, Canon EOS 700D, was used to take pictures.

## Distribution data

Fieldwork, museum collections, online biodiversity databases, and publications were used to collect distributional data. In total, 63 records were obtained from various districts of Iran over a seven-year survey period of the first author (2015–2021); 272 records were obtained from museum collections, including those at the State Museum of Natural History Karlsruhe, Germany (SMNK), the Zoological Research Museum Alexander Koenig, Germany (ZFMK), (all museum specimens were identified by the mantid specialist R. Ehrmann); 28 records were obtained from the Global Biodiversity Information Facility (GBIF; https://doi.org/10.15468/dl.y25v89), 140 from iNaturalist, including only specimens with pictures that allowed accurate species identification (confirmed by ZM); and 90 additional records from *Nasser et al. (2021)*. In total, we obtained 593 occurrence points, which were used to generate a distribution map in QGIS v. 3.22 (https://qgis.org/en/site/). Google Earth v. 9.174.0.2 (https://earth.google.com/web/)

**Table 1  Calibration and evaluation of ellipsoid models used to characterise the climatic niche of *Blepharopsis mendica*.** The table displays evaluation metrics (mean AUC, *p*-value partial ROC, omission rate), valid iterations and mean prevalence calculated in environmental ('*Prevalence on E-space*') and geographical space ('*Prevalence on G-space*'). The bold row highlights the method selected to create the final model.

| Method | Variable set | Mean AUC | Valid iterations | Partial ROC *p*-value | Omission rate | Prevalence in E-space | Prevalence in G-space |
|--------|--------------|----------|------------------|-----------------------|---------------|-----------------------|-----------------------|
| covmat | set 1 | 1.229 | 279 | <0.0001 | 0.044 | 0.933 | 0.933 |
| **mve1** | **set 1** | **1.110** | **267** | **<0.0001** | **0.044** | **0.911** | **0.911** |
| covmat | set 2 | 1.243 | 283 | <0.0001 | 0.044 | 0.924 | 0.917 |
| mve1 | set 2 | 1.205 | 87 | <0.0001 | 0.088 | 0.796 | 0.828 |
| covmat | set 3 | 1.413 | 275 | <0.0001 | 0.044 | 0.958 | 0.958 |
| mve1 | set 3 | 1.206 | 290 | <0.0001 | 0.044 | 0.942 | 0.942 |

was used to georeference specimens without coordinates based on the information present on the corresponding labels.

## Ecological niche modeling

The reduction in the number of occurrence records was achieved through spatial thinning using the R package "spThin" (*Aiello-Lammens et al., 2015*) to construct the ellipsoid niche model. To mitigate issues related to spatial autocorrelation, a minimum distance of 10 km was maintained, considering the spatial resolution of the variables (∼9.2 km at the equator) (*Kramer-Schadt et al., 2013*). This yielded a final count of 270 records, following the methodology outlined by *Cobos et al. (2018)*, which were utilized for the calibration and establishment of the final models.

Environmental data with a spatial resolution of 2.5 arc-minutes (approximately 4.6 km at the equator) were obtained from WorldClim (version 1.4, http://www.worldclim.org; *Hijmans et al., 2005*) for this study. WorldClim, based on interpolations of weather station data covering monthly precipitation, minimum and maximum temperatures from 1950 to 2000, provided 19 variables. Four variables (mean temperature of wettest quarter, mean temperature of driest quarter, precipitation of warmest quarter, precipitation of coldest quarter) were excluded due to recognized spatial inconsistencies between neighboring grid cells (*Escobar et al., 2014*; *Hijmans et al., 2005*).

Following the methodology of *Dey et al. (2021)*, three distinct environmental sets were employed to calculate the ellipsoid niche of *B. mendica*, with 'Set 1' including all 15 variables, 'Set 2' encompassing only temperature-related variables, and 'Set 3' incorporating solely precipitation-related variables. Principal component analysis (PCA) using the 'kuenm_rpca' function in the 'kuenm' package (*Cobos et al., 2019*) within RStudio 3.6.3 (*RStudio Team, 2021*; *R Core Team, 2021*) was conducted for each set. The first three components, explaining over 90% of the total variance, were retained for model calibration (refer to Table 1).

The ellipsoid models were constructed using the 'ellipsenm' package (*Cobos et al., 2020*), calibrated with the 95% pairwise confidence region for the ellipsoid, and evaluated using the 'ellipsoid_calibration' function (*Cobos et al., 2020*). Two methods were employed for

ellipsoid model creation: 'covmat,' based on the centroid and a matrix of co-variances of the variables, and 'mve1,' generating an ellipsoid minimizing volume without losing data (minimum volume ellipsoid; *Van Aelst & Rousseeuw, 2009*).

Model selection relied on statistical significance (partial ROC; *Peterson, Papeş & Soberón, 2008*), with the proportion of testing data in suitable areas and prediction of unsuitable areas determined by omission rates (E = 5%; *Anderson, Lewc & Townsend, 2003*) and prevalence (*i.e.,* proportion of space identified as suitable for the species; *Cobos et al., 2020*). Partial ROC metric calculations involved 500 bootstrap iterations, with 50% of testing data used in each iteration and 5% testing data error due to uncertainty. Prevalence was calculated in both geographical and environmental spaces, considering only pixels with distinct combinations of all variable values (*Cobos et al., 2020*; *Nuñez-Penichet et al., 2021*).

The calibration area, encompassing regions accessible to the species (*Barve et al., 2011*), featured a 50 km buffer from the occurrence records used in the models. The buffer size was determined based on observations of this species in its natural habitat, especially males with efficient wings flying to locate females for mating.

Final parameters were selected based on the best-evaluated models, and ten replicates with bootstrapped subsamples, each comprising 75% of the data and generated by excluding one occurrence record at a time, were used to create the final models. The ecological niche and suitability levels of *B. mendica* in geographical space were visualized, with binarization using a suitability threshold to exclude the 5% of data with the most extreme values. Visualization of results was carried out using QGIS v.3.10 (*QGIS Development Team, 2022*).

## Phylogeographic analyses

Mesocoxal muscle tissue from 15 preserved *B. mendica* specimens was stored in 96% ethanol. Genomic DNA was extracted using the E.N.Z.A.® Tissue DNA Kit protocol designed for animal tissue. We specifically targeted the barcoding region of the cytochrome c oxidase I (COI) gene, with a length of 658 base pairs, for amplification and sequencing. The primer sequences used were LepF1 (5′ATTCAACCAATCATAAAGATATTGG-3′) and LepR1 (5′TAAACTTCTGGATGTCCAAAAAATCA-3′), as previously described by *Hebert et al. (2004)*. Polymerase chain reaction (PCR) was conducted on a SENSQUEST Lab Cycler, with the following thermal conditions: initial denaturation at 95 °C for 5 min, followed by 38 cycles of denaturation at 95 °C for 30 s, annealing at 49 °C for 90 s, extension at 72 °C for 60 s, and a final extension at 68 °C for 30 min. Gel electrophoresis was used to confirm proper amplification and check for contaminations. The resulting PCR products were purified using Thermo Scientific Exonuclease I and the FastAP Thermosensitive Alkaline Phosphatase Clean-up Kit. Sequencing was performed at Macrogen Europe, ensuring adequate overlap with adjacent regions for sequence accuracy. Geneious R10 (https://www.geneious.com) was employed for nucleotide editing and contig assembly. A multiple sequence alignment was carried out using Bioedit 7.2.5 (*Hall, 1999*) and was subsequently converted into Fasta and Nexus formats for various analysis programs. All sequences were deposited in GenBank (https://www.ncbi.nlm.nih.gov/genbank/) with the

following accession numbers: OR588779–OR588792. In addition to these sequences, the COI barcode "MAN-00032" in BOLD was used as "Pak1". This specimen was wrongly identified in the BOLD system as a mantid from Mantidae family, while we undoubtedly identified it from the attached photo as *Blepharopsis mendica*. To visualize genetic relationships between different geographic populations, a haplotype network was constructed using the TCS network algorithm (*Clement et al., 2002*) as implemented in PopART v. 1.7.2 (*Leigh & Bryant, 2015*).

For Bayesian analysis, the Akaike Information Criterion (AIC) implemented in jModelTest v.2.1.10 was used to select the best-fitting DNA substitution models (*Guindon & Gascuel, 2003*; *Posada, 2008*). The HKY model (*Rodriguez et al., 1990*) with a significant proportion of invariant sites ($I = 0.7270$) (HKY + I) was identified by jModelTest as the best model. the analysis involved four chains, consisting of two hot and two cold chains, executed in three independent runs for 100,000,000 generations, sampling every 1,000th generation. The first 10% of generations were discarded as burn-in. We used the remaining trees with average branch lengths to create a 50% majority-rule consensus tree with the sumt option of MrBayes. TRACER (*Rambaut et al., 2018*) was used to check that analyses reached an effective sample size (ESS) over 200 in order to ensure correct chain convergence. Posterior probabilities (pp) were obtained for each clade, where pp ≥ 0.95 indicated significant support for clades. The run with the best log-likelihood score was selected. Consensus trees were visualized and rooted with *Empusa pennicornis* Pallas, 1773 as an outgroup in FigTree 1.4.2 (http://tree.bio.ed.ac.uk/software/figtree/), and edited using Inkscape vector graphics editors (ver. 1.2). *Empusa pennicornis* was chosen as the outgroup because this genus belongs to the same family (Empusidae).

Divergence time estimation was conducted using BEAST 2 v. 2.7.5 (*Bouckaert et al., 2019*). We determined the substitution model by employing jModelTest version 2.1.10. The HKY model with estimated base frequencies and gamma distribution (with 4 categories) was chosen. Due to the unavailability of fossils for *Blepharopsis* or closely related genera, we calibrated the tree using standard gene substitution rates for insects, a method also employed in prior studies (*Papadopoulou, Anastasiou & Vogler, 2010*; *Wendt et al., 2022*). Consequently, a clock rate of 0.0177 was applied based on *Papadopoulou, Anastasiou & Vogler (2010)*. To explore the potential impact of different models, we conducted two separate analyses utilizing Yule and Birth-Death tree priors. Each analysis consisted of four independent Markov Chain Monte Carlo (MCMC) runs, each running for 50 million generations and sampling trees every 5,000 generations. After discarding the initial 10% of trees as burn-in, we assessed convergence using Tracer version 1.7.1 (*Rambaut et al., 2018*). The final trees were combined using Tree Annotator v.1.10.4 and further edited using FigTree v.1.4.4 (http://tree.bio.ed.ac.uk/).

To explore the historical shifts in the geographical distribution of *B. mendica*, we employed two models for biogeographical range expansion: The Dispersal-Extinction-Cladogenesis (S-DEC) model and the Dispersal-Vicariance (S-DIVA) model, both implemented in RASP 4.3 (*Yu, Blair & He, 2020*). The input data for this analysis consisted of an ultrametric tree generated using BEAST v. 2.7.5. To enhance the precision of our
analysis, we removed the outgroup from the tree using a feature provided by the RASP software.

We delineated seven geographical regions based on our knowledge of the current distribution of the species: (A) southern and central Iran, (B) Pakistan, (C) Lebanon, (D) Tunisia, (E) Morocco, (F) Canary Islands, and (G) Oman.

To account for uncertainties stemming from the tree's structure, we incorporated all trees sampled from BEAST analyses, excluding the initial 500 trees. In the S-DIVA analysis, we selected the "Allow Reconstruction" feature, which permitted a maximum of 100 reconstructions employing three random steps. Subsequently, we conducted up to 1,000 reconstructions to obtain the final tree. Each node in the analysis has attributed the potential for up to four distinct areas.

The results of the most suitable S-DIVA reconstructions were then summarized by utilizing the pruned maximum-clade-credibility tree derived from our Bayesian phylogenetic analysis. In the S-DEC analysis, we assumed equal probabilities of dispersal between areas, and all values in the dispersal constraint matrix were set to 1, considering four as the maximum number of areas.

# RESULTS

## Field observations

**Life cycle.** Our research in the field indicates that *B. mendica* is an overwintering univoltine species. Thus, the nymphs emerge in summer (late July), as we have only found the first instar nymphs from late July to early August in their natural habitat, and they continue to grow throughout the season. Then, the larvae overwinter in the last instar (five records of living nymphs during winter from last week of October to first week of February, all of them found immobile at or in the lower parts of the dried bushes) and become adults in spring (first sightings of adults; males first week of May; females second week of May). Regarding oviposition, females began to lay their oothecae in June, as they often mate within two weeks after reaching adulthood and typically lay their first ootheca within one week after mating. However, it is important to note that oviposition timing can vary depending on various factors such as temperature, humidity, and food availability. This trend has been observed however with small differences in the distribution of this species from western North Africa to the far Middle East.

**Ootheca.** Three oothecae of *B. mendica* were collected from branches of trees or bushes during June and July 2020 (Figs. 2D, 2E). They were already hatched when collected which could be recognized by the presence of white eclosion sack-like structures in the emergence area. The eggs in this species are arranged vertically in a row next to each other as was observed by dissecting the field-collected oothecae dorsally (Fig. 2C).

**Defense behavior.** The first author observed two different responses to disturbance in this species during field surveys. Either individual flew away when disruption happened, or they extended their wings broadly and lifted their forelegs (Figs. 3A, 3B). Additionally, one female made an odd menacing gesture (Fig. 3C).

**Habitat and hosting plants.** All individuals found in the field were encountered in more or less xeric areas, with scarce vegetation composed of both herbaceous vegetation

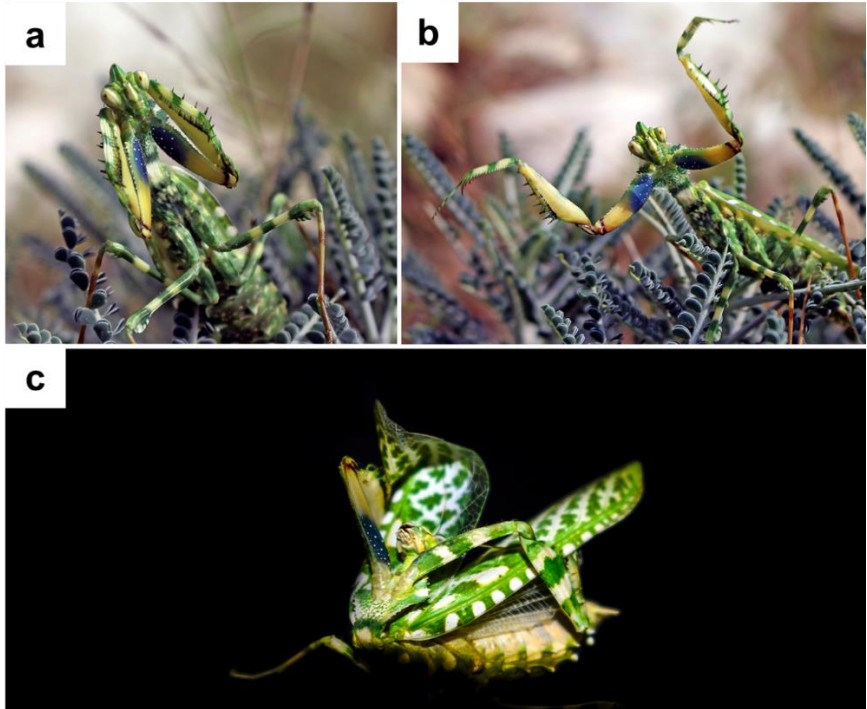

**Figure 3** *Blepharopsis mendica* threatened display: (A, B) the two commonly observed types of display, and (C) an extreme display that was observed for the first time.

and spiny bushes (Fig. 4). All specimens were found in Iran sitting on thorny bushes like *Prosopis* spec. (Fig. 5A), *Alhagi* spec. (Fig. 5B), and *Astragalus* spec. (Fig. 5C), as well as *Tamarix* spec. (Fig. 5D). Similar vegetation patterns were observed also in Morocco and Tunisia (Figs. 5E and 5F). Due to their coloring, *B. mendica* individuals are particularly suited for mimicking leaves, and prickly or dry plants, *i.e.,* the typical flora of semi-deserts (Fig. 1).

## Laboratory rearing
### Development of immatures and their color changes
One ootheca was laid in the first week of June by a female collected from the xeric shrublands of Bushehr province (N27°50′37.3″; E52°03′51.9″). This ootheca was 18 mm long, 12 mm high, and six mm wide (Fig. 2C). It had a globular shape and, as mostly in this species, a very soft texture, completely covered with a layer of spongious material, white in color at the time of laying (Fig. 2A). After one day, the color turned into a creamy color. In total, 45 nymphs hatched from the ootheca's top rim all at once after five weeks (34 days, in the second week of May).

Twenty-eight individuals (11 males, 17 females) of the 45 emerged nymphs completed their life cycle. Twelve did not reach the second instar and died possibly due to poor molting. Five died during the second and third instar. The time needed from hatching to

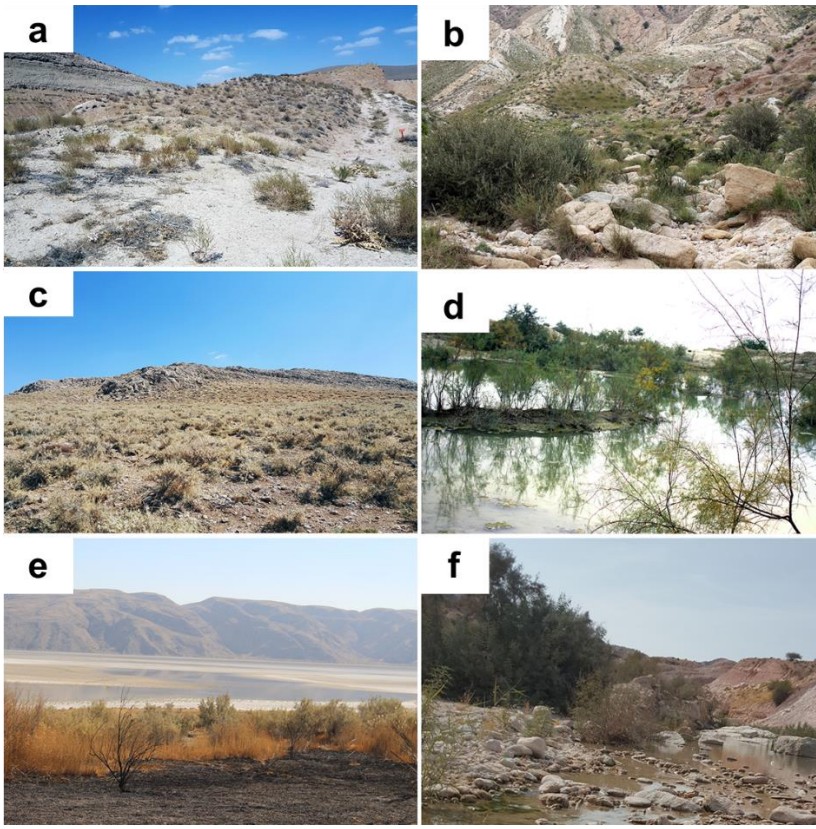

**Figure 4** **Habitat of *Blepharopsis mendica* in Iran.** (A) Abadan, Khozestan province (30.569N, 48.900E); (B) Tombak, Bushehr province (27.735N, 52.202E); (C) Kohmare Sorkhi, Fars province (29.386N, 52.177E); (D) Kangan, Busheher province (27.843N, 52.064E) ; (E) Hajiabad, Hormozgan province (28.290N, 55.887E); and (F) Salafchegan, Qom province (34.471N, 50.442E).

adulthood on average was 18 weeks (130 days) (Table 2). While most nymphs became adults after six (all males) or seven molts (most females), four females required eight molts.

The first instar had a distinct color pattern on the thorax and legs, with mostly dark brown and some white and black stripes (Fig. 1C). The color changed from light brown to creamy or white from the second instar to subadult (Figs. 1D, 1E), and the adults' color ranged from bluish green to grass green (Figs. 1A, 1B). We also observed color changes in adult specimens under laboratory conditions. Thus, three adults first appeared ochre-brown or reddish, but after some days their thoraxes became reddish, their wings greenish, and some other body parts reddish brown (Fig. 1B). The last larval instar had a longer lifespan than the others (Table 2). Overwintering of nymphs explains the long duration of the last instar since it seems that the last instar nymph will undergo a diapause process during winter (Table 2).

### Adult longevity
The mean adult longevity in the lab condition of *B. mendica* at 25 °C ± 2 was 118 days (±6 SD) for females (range: 100–124 days), and 46 days (±5 SD) for males (range: 39–55 days).

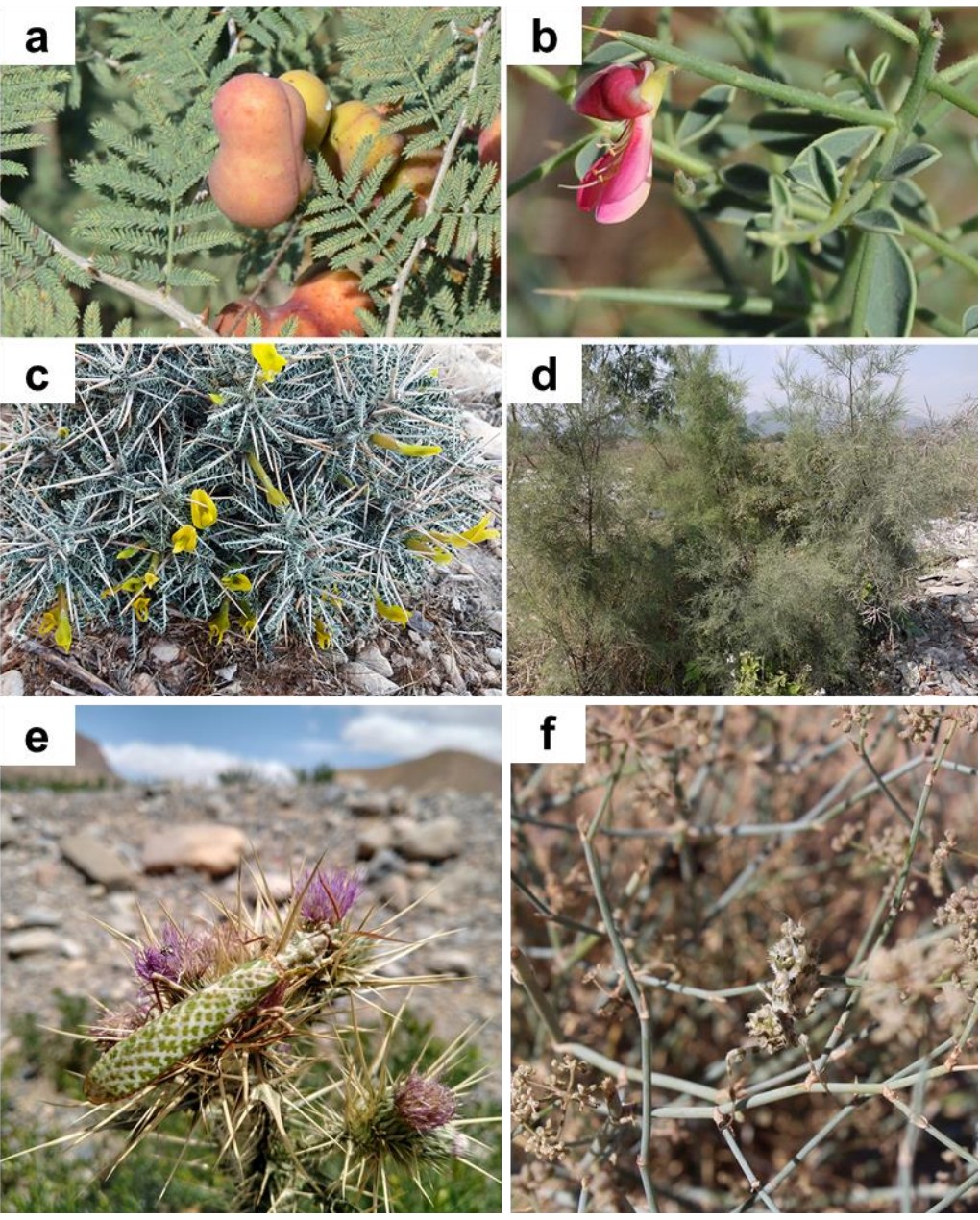

**Figure 5** **Plants on which *Blepharopsis mendica* was observed and laid oothecae.** (A) *Prosopis* spec. (Fabaceae), (B) *Alhagi* spec. (Fabaceae), (C) *Astragalus* spec. (Fabaceae), (D) *Tamarix* spec. (Tamaricaceae), (E) *Ptilostemon* spec. (Asteraceae), and (F) *Deverra* spec. (Apiaceae). (A–D) Iran, (E) Morocco, (F) Tunisia.

**Table 2  Mean duration (in days) ± standard deviation of each of the respective life stages of**
*Blepharopsis mendica* **and differences of male and female development under laboratory conditions.**

| Life stage | Males | Females |
|---|---|---|
| 1st instar | 11.5 ± 1.3 | 11.3 ± 1.3 |
| 2nd instar | 12.5 ± 1.2 | 12.5 ± 1.0 |
| 3rd instar | 11.0 ± 1.2 | 11.1 ± 0.9 |
| 4th instar | 12.0 ± 1.2 | 12.4 ± 1.4 |
| 5th instar | 11.7 ± 1.1 | 11.6 ± 1.4 |
| 6th instar | 92.5 ± 2.9 | 12.8 ± 2.6 |
| 7th instar | N/A | 26.0 ± 7.0 |
| 8th instar | N/A | 105.3 ± 3.3 |
| Total nymphal period | 151.1 ± 7.2 | 193.3 ± 9.2 |
| Adult longevity | 46.6 ± 4.7 | 118.4 ± 6.4 |
| Period from hatch to death | 197.8 ± 10.1 | 311.7 ± 9.3 |

SD refers to standard deviation. The $t$-test recovered a statistically significant difference
($P < 0.001$) when comparing sexes (Table S3). The average total life cycle was 216 days
($\pm 9$ SD) for females, and 132 days ($\pm 7$ SD) for males ($P < 0.27$) (Table S4).

*Oviposition*

To test for parthenogenesis, five of the 17 females who reached adulthood were not
mated. Three of these produced three unfertilized oothecae, none of which hatched. The
12 remaining females were joined with the males in a separate terrarium for mating.
Eight females successfully mated and produced a total of 11 oothecae, *i.e.,* four laid one
ootheca, two laid two, and one laid three. Only four of these oothecae hatched. No sexual
cannibalism was observed during this study.

There were no observable physical differences or deformations between the unfertilized,
unhatched, and hatched oothecae (Table 3). However, the number of eggs per ootheca
varied depending on the type and size of the ootheca (Table 3). The average number of
eggs per ootheca was higher in the hatched (mean: $43.7 \pm 7.2$ SD) and unhatched (mean:
$31.8 \pm 2.4$ SD) oothecae compared to the unfertilized ones (mean: $18.0 \pm 2.9$ SD) (Table 3).
ANOVA tests indicated significant differences among the three groups for all characteristics
(*i.e.,* weight, length, width, and number of eggs), except for height. There is also a significant
difference in weight and length between the hatched and unfertilized ootheca, as both Tukey
$p$-values are less than 0.05, but there is no significant difference in height or number of eggs.
Comparing hatched and unhatched oothecae revealed significant differences in weight,
length, and number of eggs; however, there was no significant difference in width or height
(Table 4).

## Distribution and ecological niche modelling

*B. mendica* is largely associated with dry grasslands, desert and semi-desert, and xeric
shrublands from the Canary Islands to Pakistan (Fig. 6). Almost identical vegetation
pattern was observed in Morocco and in Tunisia where this species has been observed in
the wild. In Iran, where the distribution was poorly known prior to this study, it is also

**Table 3** Mean weight, size, incubation duration, hatching number, and the number of internal egg chambers of the various types of oothecae of *Blepharopsis mendica* reared under captive breeding conditions.

| Ootheca type | Weight [mg] | Length [mm] | Width [mm] | Height [mm] | Incubation duration [d] | Hatching No. | Number of eggs |
|---|---|---|---|---|---|---|---|
| Unfertilised | 160 ± 30 | 11.9 ± 0.4 | 2.5 ± 0.1 | 6.9 ± 0.3 | N/A | N/A | 18.0 ± 2.9 |
| Unhatched | 360 ± 30 | 20.1 ± 0.8 | 4.2 ± 0.2 | 10.7 ± 0.2 | N/A | N/A | 31.9 ± 2.4 |
| Hatched | 460 ± 20 | 28.9 ± 2.0 | 4.2 ± 0.2 | 11.0 ± 0.4 | 36.8 ± 2.9 | 42.3 ± 5.6 | 43.8 ± 7.2 |

**Notes.**
SD, standard deviation.

**Table 4** Analysis of variance (ANOVA) and associated *post hoc* Tukey $p$-value between the three types of *Blepharopsis mendica* ootheca and the various morphological parameters.

| Statistical test | Oothecae | Weight | Length | Width | Height | Number of eggs |
|---|---|---|---|---|---|---|
| ANOVA | Overall | <0.001[*] | <0.001[*] | <0.001[*] | 0.117 | <0.001[*] |
| *Post Hoc* (HSD Tukey) | Unfertilized × Unhatched | <0.001[*] | <0.001[*] | <0.001[*] | 0.03[*] | <0.001[*] |
| | Unhatched × Hatched | <0.001[*] | <0.001[*] | 0.88 | 0.76 | 0.03[*] |
| | Hatched × Unfertilized | <0.001[*] | <0.001[*] | 0.02[*] | 0.08 | 0.98 |

**Notes.**
*Significant $p$-value <0.05, marked by *.

widely distributed, only excluding the driest regions in the central and eastern parts of the country and the high mountain areas in the west. The new records from Iran are listed in Supplemental Information 1. The geographic projections of the ecological niche of *B. mendica* showed widespread climatic suitability across North Africa and southwestern Asia; lower suitability was recovered for the Sahel zone and southern Africa (Fig. 7, Fig. S1). The best fitting method to construct the climatic ellipsoids was 'mve1', with environmental set 1, containing principal components of all 15 variables; mean AUC, $p$-value of partial ROC, and omission rates were significantly better than random expectations ($P < 0.05$; Table 1). The prevalence of mean ellipsoidal models in geographical (G-space) and environmental (E-space) space was relatively high (0.912; Table 1). The complete report of ellipsoid characteristics (*e.g.*, centroid, covariance matrix, semi-axes length, *etc.*) is given in Supplemental Information 3.

## Divergence dating, biogeography, and phylogenetic analyses

COI sequences of 15 specimens of *B. mendica* revealed 12 different haplotypes (Fig. 8). Bayesian tree and haplotype network analysis of *B. mendica* identified three distinct population groups: (i) Pakistan, (ii) Maghreb from Morocco to Tunisia, including the Canary Islands, and (iii) Middle Eastern populations from Lebanon, Oman, and Iran; the latter group is subdivided into a western subgroup (iiia) in Lebanon, Oman, and Khozestan (border Iran/Irak) and an eastern one (iiib) widespread in southern and central Iran (Figs. 9 and 10). Our biogeographic analysis using S-DIVA and S-DEC models revealed a divergence of the lineage in Pakistan from the ancestor of the other groups about 1.5 million years ago. Another vicariance event separated the Maghreb populations from the remaining ones about 1.3 Mya. Less than 1 Mya, a dispersion event led to the split

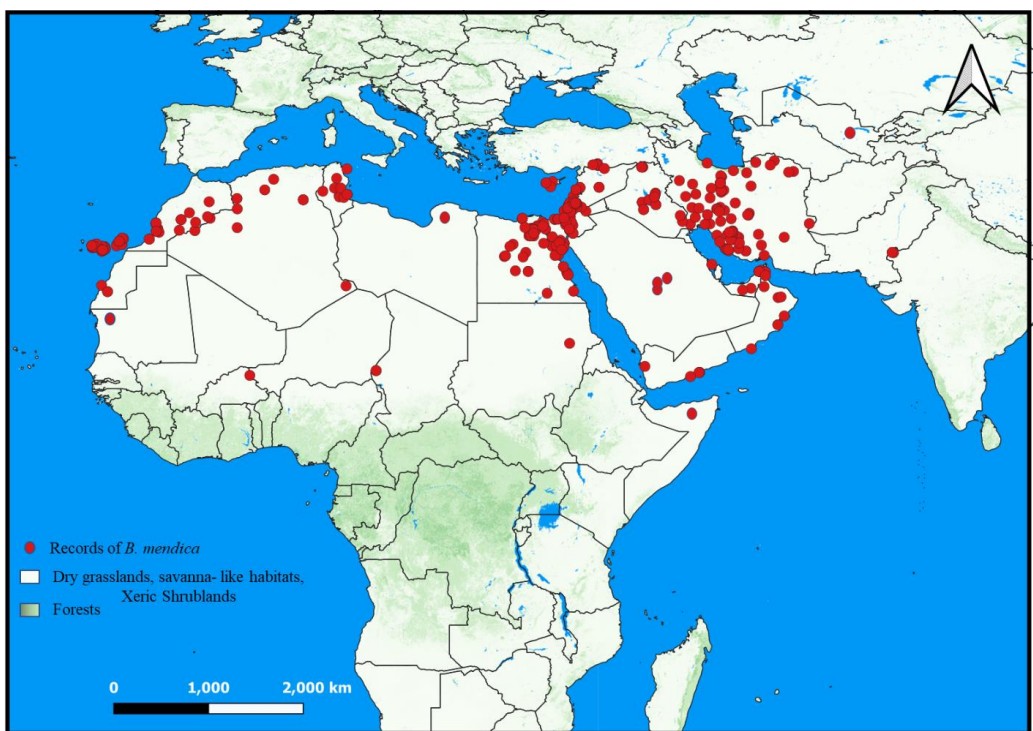

**Figure 6** Distribution of *Blepharopsis mendica* according to the available records. The prevailing habitat types are given for the distribution range (QGIS v. 3.22).

between the Middle East (Lebanon, Oman, Khozestan) and most other Iranian populations (Fig. 9). Hence, historical events, including vicariance and dispersion, played pivotal roles in shaping the genetic pattern of *B. mendica* populations.

## DISCUSSION

### Life cycle and variability in nymphal development

The life cycle of mantids is divided into two phases: the developmental period from hatching to reaching adulthood and the reproductive period as adults, which is defined by adult longevity. *Korsakoff (1934)* reared specimens of north African populations, and he recorded nine instars for females and eight instars for males of *B. mendica* from hatching to adulthood. In our study, nymphs only passed through fewer instars, *i.e.,* six for males and seven (rarely eight) for females, which is similar to *Idolomantis diabolica* (Saussure, 1869), which passes seven instars to reach adulthood (*Schwarz, Mehl & Sommerhalder, 2007*), or *Hierodula* species, which pass six to nine instars to reach adulthood (*Leong, 2009*; *Raut, Bhawane & Gaikwad, 2014*; *Mirzaee, Sadeghi & Battiston, 2022*). The variability in the number of instars in mantids might be due to different factors, such as temperature, resource availability and quality, humidity, genetics, sex, and photoperiod (*Esperk, Tammaru & Nylin, 2007*). Therefore, a higher temperature, humidity, and resource availability and quality in our study might decrease the number of molts in this species.

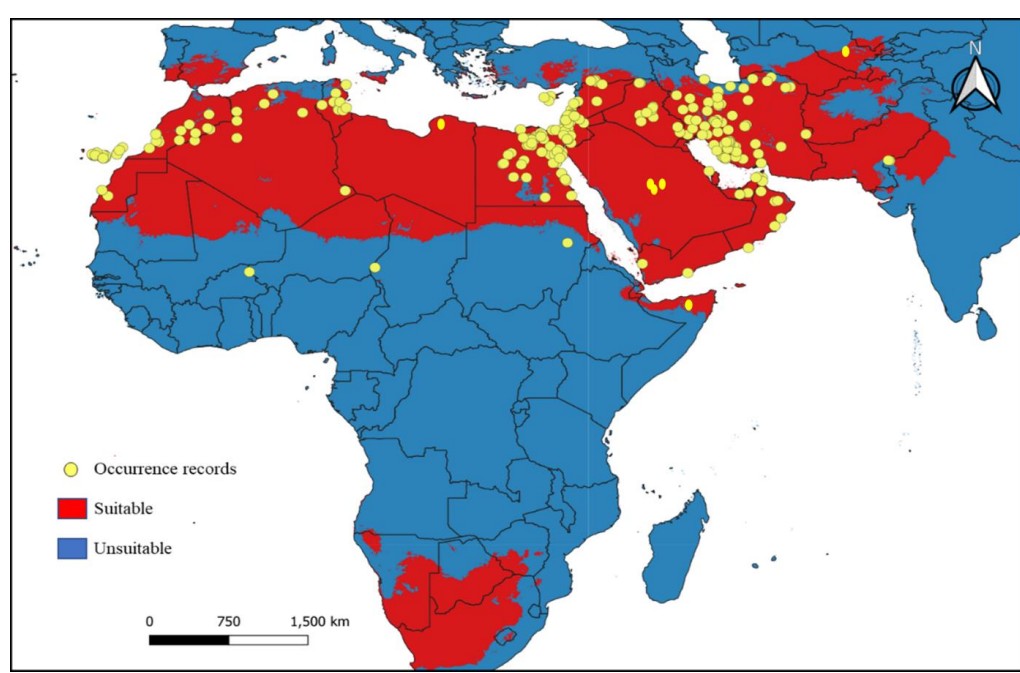

**Figure 7 Current climatic suitability for *Blepharopsis mendica*.** Map showing the threshold (E = 5%) of current climatic suitability for *Blepharopsis mendica* in its native range. Red indicates areas of high climatic suitability, whereas blue represents areas with lower climatic suitability.

The higher temperatures used to rear the nymphs of *B. mendica* in our study (33–35 °C), in comparison to Korsakoff's study (27 °C), could have accelerated the developmental rate of the specimens, resulting in fewer instars being needed to reach adulthood. Similarly, if the quality and availability of food were different between the two studies, this could have also influenced the developmental rate and the number of instars required for the mantids to reach adulthood. In Korsakoff's study specimens were fed by rose moth caterpillars but in our study, we used mealworms, flies, and grasshoppers. Additionally, differences in the genetic background and sex of the mantids used in the two studies could also have contributed to the differences in the number of instars *e.g.*, the mantids used in this study were from Iran and the mantids Korsakoff used in his study were from North Africa.

As in our study, *Maxwell (2014a)* also observed a similar variation in the number of instars in *Stagmomantis limbata* bred in captivity, with 64% of nymphs requiring six, and 36% requiring seven instars. He considered this variation in the number of instars as a "bet-hedging" strategy used by females to produce variation in development among siblings (*Maxwell, 2014b*). It thus might be a survival strategy, for mantid species in general and for such species living in extreme and often largely unpredictable habitats like *B. mendica* in particular, because sisters hatching together will enter the reproductive phase at different points in time. This is increasing the chance that at least some females are reproductive in a suitable time window, hence safeguarding the survival of regional populations of the respective species.

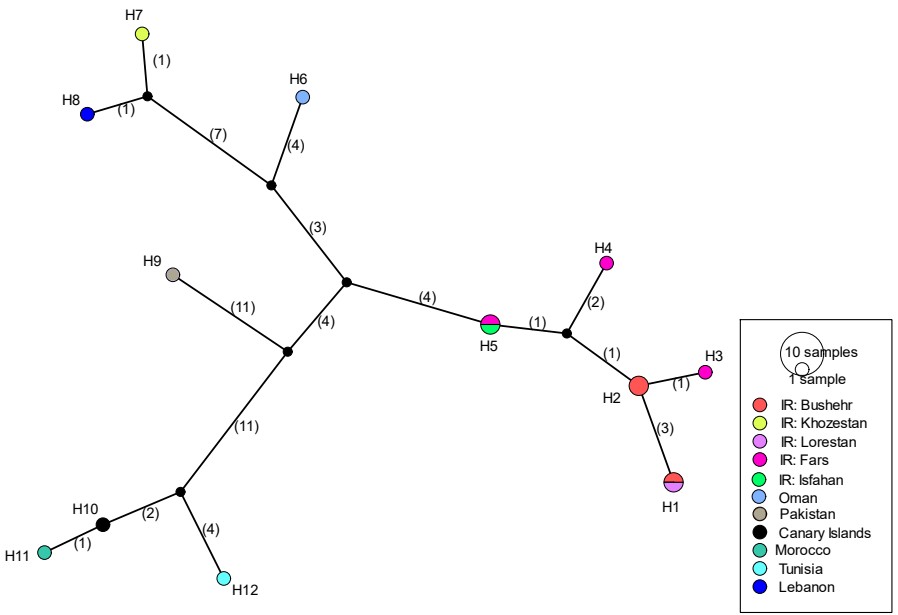

**Figure 8** **Haplotype network of _Blepharopsis mendica_ based on the COI gene fragment.** Circles sizes are proportional to haplotype frequency, black dots represent missing haplotypes. Colours refer to different localities.

## Adaptations of the incubation time of oothecae, and nymphal overwintering

The incubation duration of oothecae in Mantodea often is species-specific, but can also be influenced by the environment. Therefore, it is important to consider the impact of environmental conditions when studying the developmental biology of any species (_Greyvenstein, Du Plessis & Van den Berg, 2022_). It seems that temperature, particularly daily maximum temperature, is the key factor for hatching in different mantid species such as _A. spallanzania_ (Rossi, 1792) (_Battiston & Galliani, 2011_). Various mantis species employ distinct strategies for overwintering and development, demonstrating their ability to adapt to diverse environmental conditions. These differences may be influenced by specific genetic factors, potentially resulting in different life cycles even when multiple species share the same habitat. Overwintering strategies in Mantodea can be different between different genera but little information regarding these strategies is available for this group of insects. Some Mantidae genera, for example, _Hierodula_ Burmeister, 1838 and _Mantis_ Linneus, 1758, go into a facultative diapause phase during the ootheca stage (_Ramsay, 1984_; _Schwarz, Keller & Berger, 2017_; _Mirzaee, Sadeghi & Battiston, 2022_). However, some other species in different genera like southern species of _Ameles_ Burmeister, 1838, _Empusa_ Illiger, 1798, and _Severinia_ Finot, 1902, have the strategy to overwinter as nymphs (_Battiston & Galliani, 2011_; _Shcherbakov & Govorov, 2021_). In our study, the last nymphal instar of _B. mendica_ lasts longer than the previous ones (as shown in Table 2). This developmental pattern is also seen as an adaptation strategy to survive overwintering as a nymph.

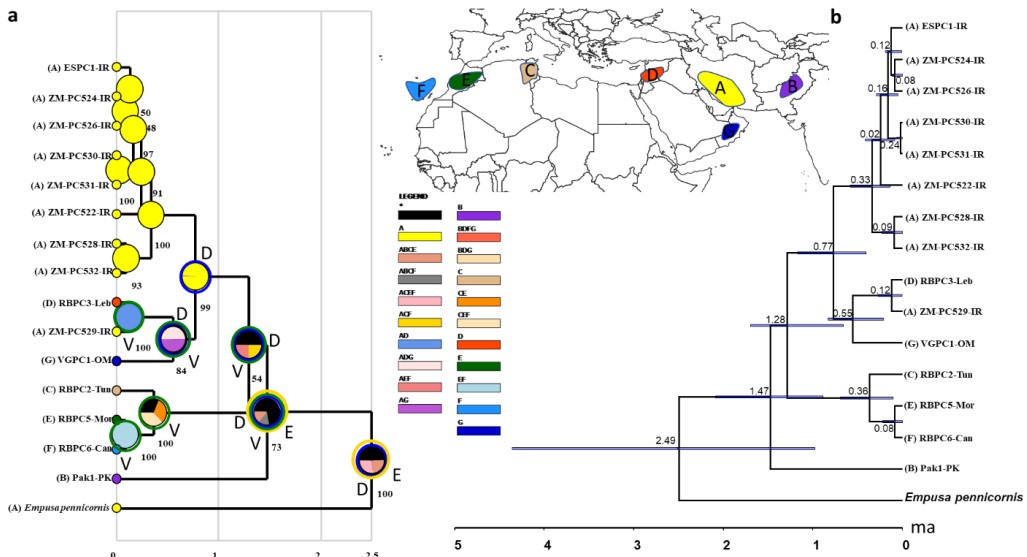

**Figure 9** **Divergence time and biogeography of *Blepharopsis mendica*.** (A) Ancestral range estimation of *Blepharopsis mendica*. The biogeographic reconstruction of RASP S-DIVA biogeographical analysis models (max. number of areas = 4). The pie charts indicate alternative ancestral geographical ranges and their probabilities. Numbers besides pie charts are probability values for nodes. Species were assigned to the five distribution areas A to E as illustrated on the inset map and the respective tip ranges (coloured squares with letter codes at tips). The legend below the inset map displays the colour codes for each area, including the area combinations as retrieved in the analysis. D = dispersal, blue circles around pie charts, V = vicariance, green circles around pie charts, and E = extinct, orange circles around pie charts. b) Phylogeny and diversification of *B. mendica* based on a COI tree constructed in *Beast. 95% highest posterior probabilities are shown with blue bars. IR: Iran, Leb: Lebanon, PK: Pakistan, Can: Canary Islands, Mor: Morocco, Tun: Tunisia, OM: Oman.

In our study, the average incubation period for oothecae of *B. mendica* was 36.8 days (±2.9 SD). This is similar to other members of the Empusidae family, such as *Empusa pennata* (Thunberg, 1815) (28 days) (*Bischoff, Heßler & Meyer, 2001*), *Idolomantis diabolica* (42–72 days) (*Schwarz, Mehl & Sommerhalder, 2007*), or Mantidae family members such as *Orthodera ministralis* (30.9 days), and *Hierodula ventralis* (25 days) (*Suckling, 1984*; *Raut, Bhawane & Gaikwad, 2014*; *Mirzaee, Sadeghi & Battiston, 2022*). However, shorter (*e.g.*, 16 days for *Ephestiasula pictipes*; Hymenopodidae) and much longer incubation periods (*e.g.*, 142–209 days for *Stagmomantis limbata*; Mantidae) also exist (*Robert, 1937*; *Vanitha et al., 2016*). Thus, the adaptation strategy for the incubation period can vary across different species. Even among species with similar incubation periods, the strategies used can be different. For instance, females of *Hierodula tenuidentata* lay their oothecae in late autumn, which then undergo a dormant process during winter, and egg development begins when temperatures become suitable; the same also applies to *Mantis religiosa* and *Sphodromantis viridis* (*Kaltenbach, 1963*; *Berg, Schwarz & Mehl, 2011*, *Mirzaee, Sadeghi & Battiston, 2022*; *Raut & Gaikwad, 2016*). On the other hand, females of *B. mendica* and *Empusa* spp. lay their ootheca in spring so that it is the nymphs that needs to overwinter having a hereon adapted life history.

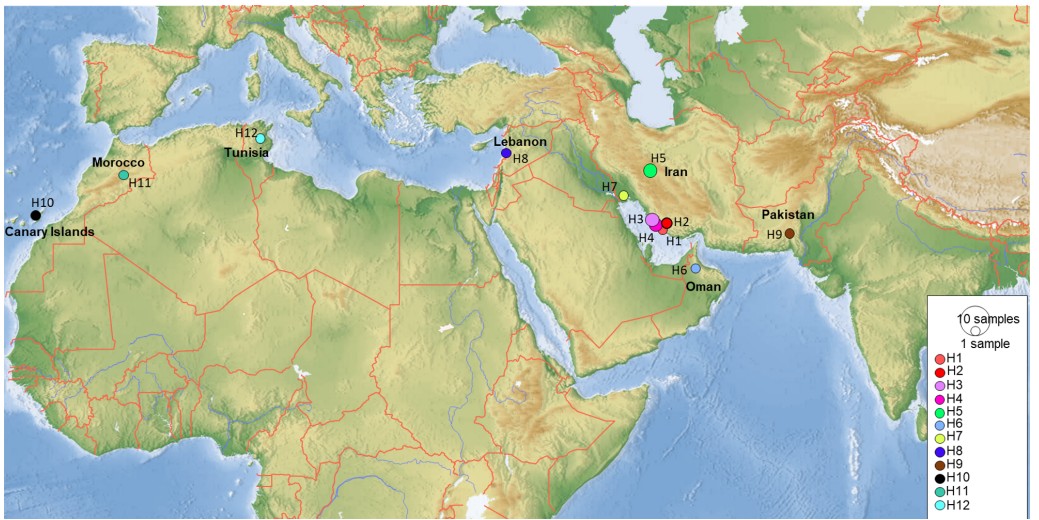

**Figure 10** Map showing the distribution of the 12 COI haplotypes obtained for *Blepharopsis mendica* genetic samples.

Having winter dormancy and a short incubation time of the ootheca afterwards seems to be an appropriate strategy in sub-Mediterranean and temperate climates where winters are too cold for nymphs of many species to overwinter successfully. In these regions, only oothecae can endure winter months. The overwintering of nymphs represents an older strategy, observed not only in these climates but also in tropical regions with a dry season. However, colonization of higher latitudes is only possible for species having oothecae (or rarely nymphs) adapted to cold conditions. In contrast, temperate species require longer incubation periods or even dormancy, especially when the egg is in the overwintering stage. For species living in regions with an arid summer and a mild winter climate, an intermediate incubation time might be the most suitable adaptation. This is because egg maturation takes place during the hottest and driest time of the year when there is limited food supply. Then, the larvae hatch with the first autumn rains, and reproduction in the following year will end when the living conditions become unfavorable (*Robert, 1937*; *Vanitha et al., 2016*; *Raut & Gaikwad, 2016*; *Mirzaee, Sadeghi & Battiston, 2022*).

## Color polymorphisms, variability as an adaptive defense mechanism, and startling displays

Coloration can be influenced by genetic factors and environmental conditions (*Okay, 1953*; *James, 1944*). The different colors of different instars and the color changes of *B. mendica* therefore might be interpreted as an adaptational defense according to the respective environmental conditions and the ability of the species to camouflage and thus avoid predators. A similar developmental strategy was also observed for *Mantis religiosa* often changing its color from brownish to green along its larval development (*Battiston & Fontana, 2010*; *Berg, Schwarz & Mehl, 2011*). Under artificial conditions, some adult brown *M. religiosa* females showed an incomplete but clearly visible variation toward green tones in some body parts, even when no green components were available in the cages. Even

more, *B. mendica* is able to change its color in the adult stage without using the renovation processes of a molt. This latter phenomenon is also known for *Miomantis caffra* (*Ramsay, 1990*) and *M. religiosa* (*Okay, 1953*); the mechanisms behind it are still unknown.

Startle displays in praying mantises are intricate performances designed to deter or confuse predators, significantly enhancing biological fitness. These displays involve a striking combination of movements, colors, and sounds. Despite their complexity, our comprehension of the underlying mechanisms, evolutionary connections, and the specific conditions influencing the performance and evolution of these displays remains incomplete (*Vidal-García et al., 2020*). In our study, individuals of *B. mendica* exhibited two distinct startle displays. Notably, one of these displays involved an unusual menacing gesture (Fig. 3C), which appears to have never been observed in this or any other Mantodea species. In general, various display behaviors exist in different Mantodea species, varying in their orientation and dynamics. For example, Devil's flower mantis (*Idolomantis diabolica*) individuals engage in repetitive back-and-forth actions accompanied by sound, while the European mantis (*Mantis religiosa*) repeatedly stridulates using its wings (*Edmunds, 1972*; *Schwarz, Mehl & Sommerhalder, 2007*). Interestingly, in our study, a startling display similar to that observed in *Idolomantis diabolica* was noted in *B. mendica* individuals, with the distinction that the specimens in our study did not display a sound threat. However, Christian Schwartz's observations (personal communication) indicated sound displays also for *B. mendica* specimens. The diversity of startle displays is evident. Thus, some species are capable of giving either a 'frontal' or a 'lateral' display depending on the predator's angle of attack, while others exhibit only lateral or frontal displays. In our study, *B. mendica* specimens displayed both frontal and lateral presentations. The positioning of forelegs during the display varied from flexed to outstretched (Fig. 3), adding to the complexity of these signals. Such extreme complexity combined with intra- and interspecies variability presents exciting avenues for future research, underscoring praying mantises as a model group to deepen our understanding of the evolution of startle displays (*Vidal-García et al., 2020*).

## Characteristics of ootheca and parasitization

Various factors, including male presence, temperature, humidity, food availability, and genetics, affect the size, color, and structure of oothecae (*Robert, 1937*); Breland and Dobson 1947; (*Hurd et al., 1995*). Mantid oothecae are consumed by certain beetles (*Orphinus* spp. *Attagenus* spec. *Phradonoma* spec.; Dermestidae) and parasitized by wasps (*e.g.*, *Anastatus* spp.; Eupelmidae, *Podagrion* spp.; Torymidae) (*Kershaw, 1910*; *Hawkeswood, 2003*; *Bolu & Ozaslan, 2015*; *Mirzaee, Lotfalizadeh & Sadeghi, 2021*; *Mirzaee et al., 2022*). These factors have a significant impact on not only the appearance of oothecae but also their survival rates and hatching, and therefore the population dynamics of mantids in their natural habitats. *Korsakoff (1934)* discovered that the chalcidoid wasp *Podagrion* spec. parasitized the oothecae of *B. mendica* that he collected from North Africa, with more parasitoids than *B. mendica* nymphs emerging. In our study, none of the oothecae were parasitized due to laboratory conditions, but further research is necessary to identify the species of beetles or wasps preying or parasitizing on *B. mendica* oothecae in the wild.

## Distribution and ecological biogeography

Our climate suitability model recovered suitable areas that well reflect the known distribution of *B. mendica* (Figs. 7 and 8), ranging from the Maghreb in the west to the Middle East as far east as Pakistan and the driest parts of western India. Hence, high climatic suitability was exclusively recovered in hot and dry regions (Fig. S1). However, *Akhmedov & Kholmatov (2019)* refer to specimens labeled as originating from Uzbekistan (*i.e.,* Kreizberg's collections along the Nurata ridge during 1991–1993, housed in the museum of the Institute of Zoology of the Academy of Sciences of the Republic of Uzbekistan, Tashkent). Interestingly, our climate model identified a climatically suitable area restricted to this particular part of Uzbekistan. Despite the remoteness of this occurrence in Nurata ridge from all other data points known, the plausibility of this record therefore is high. If being searched for again, a future rediscovery is likely. A recent study conducted by *Nasser et al. (2021)* analyzing *B. mendica* in Egypt equally found that temperature-related variables but also low altitude were the factors most significantly contributing to the climatic niche model. In their study in contrast to our work, however, precipitation-related variables had a relatively small influence because their focal area was rather uniform in terms of having rather low precipitation rates.

However, a real geographic split into two major groups is possible, one Maghreb group, largely distributed in north-western Africa, and one group around the Arabian Peninsula, ranging from Egypt *via* Israel, Iraq, and Iran to Yemen. The formation of these two major distinct geographic groups in *B. mendica* might be the result of a combination of historical and extant environmental factors shaping the distribution and genetic makeup of the species over time. Thus, these two major groups originating from one common ancestral population might have been separated by a physical barrier, such as temporally existing stretches of extreme desert where no oases or vegetated wadis were available preventing gene flow among these groups. Over time, genetic differences might have accumulated through genetic drift or natural selection, leading to the formation of these two major distinct gene pools. Environmental factors, such as differences in climate and vegetation (*Mulligan, Keulertz & McKee, 2017*), also might have played a role in shaping the distribution of *B. mendica*. For example, the driest parts of the Arabian Peninsula may not provide suitable habitats for the species, whereas some oases or vegetated wadis and the more humid areas along its coastlines as well as the southern Maghreb region may provide more favorable conditions.

Suitable climatic conditions were also recovered in parts of southern Africa. Thus far, however, the true absence of *B. mendica* in this region might be due to the interspersed, geographically rather large regions whose climatic conditions permanently have been completely unsuitable (*i.e.,* tropical forests of central and eastern Africa; Fig. S1), in combination with the limited dispersal capability of *B. mendica*.

Despite some known occurrences in the Sahel zone, our model recovered only marginally suitable climatic conditions for this region. Consequently, three plausible explanations arise for these Sahel zone records: (1) the species may be exceptionally rare in this zone, limited to areas with oases or vegetated wadis, offering only marginal living conditions; or (2) the species might be more common in the Sahel zone, and the conditions are suitable, but

the region is largely understudied for this particular species; (3) artificial introductions (intended or unintended, *e.g.*, *via* trucks) still cannot be ruled out. Additionally, there is a noticeable gap between the populations in Pakistan and Iran. This gap could be genuine, reflecting an actual absence, or it could be attributed to insufficient study and undersampling in this area, which requires further investigation for clarification. Therefore, additional studies in the Sahel and other understudied regions are mandatory to address and resolve these open questions.

The only major region where *B. mendica* was frequently observed in areas not indicated as suitable by our model is the mountainous parts of northern Iran. We believe that this is due to the complex climatic structuring of this area with very heterogeneous microclimatic conditions (*Heshmati, 2007*). The hot and dry conditions needed by *B. mendica* are mostly restricted to relatively small pockets in the landscape, such as deep valleys, so the species is occurring rather locally. As the climate in most parts of these landscapes is unsuitable for *B. mendica* at the grit level, our model likely was unable to detect these small-scale pocket-like occurrences. This model confirms the general conservation assessment of this species (*Battiston, 2016*) which hypothesized the existence of diminutive and fragmented local populations (as in oases all across the Sahara and Saudi-Arabia) within the extensive distribution range of *B. mendica*.

### Divergence dating and phylogeographic analyses

Our study also provides insights into the evolutionary and biogeographic history of *B. mendica*. The distinct genetic lineages identified in Pakistan, north-western Africa (Morocco, Tunisia, Canary Islands), the Middle East (Lebanon, Oman, Iran-Iraq border, most likely Egypt), and Iran (south and central regions) reflect that the species exhibits a preference for specific environmental conditions which remain similar despite the species' extensive distribution. In addition, interchanging periods of humidity (with habitat regressions) and aridity (with habitat expansions) might have shaped the observed haplotype distribution, and enabled the species to colonize remote oases and wadis in the Sahara and areas as far east as the Thar desert (Figs. 9 and 10).

The separation of *B. mendica* from other Empusid mantids might have occurred around 2.5 mya suggesting that this species has evolved independently from other Empusid mantids all along the Pleistocene. Whether *Idolomantis diabolica* (Svenson et al. 2015 based on genetic analyses) is the sister group of Blepharodinae with its two monotypic genera *Blepharodes* Bolivar, 1890 and *Blepharopsis* Rehn, 1902 or belongs to the Empusinae (*Wieland, 2013*; *Schwarz & Roy, 2019*, based on morphological analyses) is still debated. However, our phylogenetic analyses (Fig. S2) did not deliver information resolving these conflicting interpretations as it reveals a polytomy of these three taxonomic entities. The subsequent divergence of the Pakistan lineage from the remaining populations around 1.5 mya may have been influenced by geographic barriers or environmental changes, maybe going along with the general aridification alongside the mid-Pleistocene Transition (1.2–0.8 mya) (*Thunell, 1979*; *Bertoldi, Rio & Thunell, 1989*; *Berends et al., 2021*).

The separation of the Maghreb lineage a little later, *i.e.*, around 1.3 mya, likely also resulted from vicariance that again could have been triggered by the mid-Pleistocene

Transition's aridification (*Berends et al., 2021*) indicating that climate-driven geographical isolation might have played an important role in the differentiation of *B. mendica*. Less than one mya and hence at the end of the mid-Pleistocene Transition, a dispersal event out of Iran (detected by our RASP analysis) was responsible for the colonization of the Arabian Peninsula or the Middle East with subsequent vicariance and differentiation among these three regions. The arid Pleistocene conditions in the Maghreb region prevailing during most of the last 0.5 my might also be responsible for vicariance between its eastern and western regions for *B. mendica* assumed 360,000 years ago, an often-observed fact in this region, but mostly with considerably higher vicariance age (*Husemann et al., 2014*) or it might also be probable that the humid periods ("green Sahara") drove the species into arid refugia. The colonization of the Canary Islands is a rather recent event dated by our molecular clock to 80,000 years before the present, and hence immediately before the true onset of the Würm glaciation (*Ampferer, 1925*).

## CONCLUSION

This study adds information on the little-known desert mantid species *B. mendica*, including its life cycle, ootheca (egg case), defense behavior, and preferred habitat. Additionally, our climate suitability model provided important insights into the species' distribution, corroborating existing records while also pointing out areas where sampling has been limited and regions that still remain unexplored. However, to fully understand the distribution patterns with its underlying phylogeographical structures and the factors shaping the ecological niche of *B. mendica* across different geographical regions, further research, fieldwork, and validations are essential. These efforts will contribute to a more comprehensive understanding of the species distribution and its relationship with environmental factors.

## ACKNOWLEDGEMENTS

We wish to express our sincere appreciation to Christoffer Fägerström, curator at Lund Museum of Zoology, Sweden, Peters Ralph, curator at Zoological Research Museum Alexander Koenig, Germany (ZFMK), and Alexander Riedel, curator at State Museum of Natural History Karlsruhe, Germany (SMNK), for generously providing us with invaluable coordinates and data. Additionally, we would like to express our gratitude to Oscar Maioglio (World Biodiversity Association, WBA, Italy), Evgeny Shcherbakov (Lomonosov Moscow State University, Russia), and Valeriy Govorov (Charles University, Prague, Czech Republic) for providing the legs of the samples collected from Canary Islands, Fasa, Fars, Iran and Oman. We would also like to thank Hossein Abdollahi for his aid during fieldwork, and Shiraz University Genetic lab for providing *Drosophila melanogaster* in order to feed first instar nymphs.

### Funding
The authors received no funding for this work.

### Competing Interests
The authors declare there are no competing interests.

### Author Contributions
- Zohreh Mirzaee conceived and designed the experiments, performed the experiments, analyzed the data, prepared figures and/or tables, authored or reviewed drafts of the article, and approved the final draft.
- Marianna V.P. Simões conceived and designed the experiments, analyzed the data, authored or reviewed drafts of the article, and approved the final draft.
- Roberto Battiston conceived and designed the experiments, authored or reviewed drafts of the article, and approved the final draft.
- Saber Sadeghi conceived and designed the experiments, authored or reviewed drafts of the article, and approved the final draft.
- Martin Wiemers conceived and designed the experiments, authored or reviewed drafts of the article, and approved the final draft.
- Thomas Schmitt conceived and designed the experiments, authored or reviewed drafts of the article, and approved the final draft.

### DNA Deposition
The following information was supplied regarding the deposition of DNA sequences:
 All sequences are available at GenBank: OR588779–OR588792.

### Data Availability
 The information regarding the samples used in this study, metadata obtained from Ellipsoid niche model, and a list of the new records are available in the Supplementary File.

### Supplemental Information
Supplemental information for this article can be found online at http://dx.doi.org/10.7717/peerj.16814#supplemental-information.

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
