# Peer review of "Biology, ecology, and biogeography of eremic praying mantis Blepharopsis mendica (Insecta: Mantodea)"

_PeerJ, doi:10.7717/peerj.16814_

## Round 0.1 · original submission · Minor Revisions

Dear Authors

Your submitted manuscript is well-written and presents relevant information about Blepharopsis mendica. However, to enhance your results, I recommend you provide additional results in the modeling section.

Reviewer 1 ·

Basic reporting

The paper titled "Biology, Ecology, and Biogeography of the Enigmatic Desert Praying Mantis Blepharopsis mendica" provides valuable insights into the life cycle, behavior, habitat, and distribution of B. mendica. The authors have conducted field observations, gathered distribution data, and performed phylogeographic analyses to support their findings. Overall, the manuscript is well-written and presents relevant information. However, I recommend that the authors provide additional results in the modeling section to further enhance the study.

The manuscript is generally well-written, but there are a few instances where the language could be improved for better clarity. I suggest the authors review and revise the following sections: lines 23, 77, 121, and 128. It may be beneficial for the authors to have a colleague proficient in English and familiar with the subject matter review the manuscript or consider consulting a professional editing service.

Experimental design

Modeling Section:
The modeling section of the manuscript could benefit from additional results. The authors have mentioned that ecological niche modeling was performed using ’Set 1’ included all 15 variables, ’Set 2’ included only temperature-related variables and ’Set 3’ included only precipitation-related variables., but they have not provided the outcomes or discussed the suitability of the models used. It would be valuable to include details on the modeling techniques employed, the evaluation metrics used to assess model performance, and the resulting predictions or projections. This additional information will strengthen the study's findings and provide a more comprehensive understanding of the species' distribution patterns. Also, the provided map is with not accepted color with me and why they use present/absent method for this map; I encourage authors to provide continuance map with percentage of suitability.

Validity of the findings

Significance and Impact:
While the study provides valuable insights into the biology, ecology, and biogeography of B. mendica, the authors should explicitly discuss the significance and impact of their findings. They should address how their research fills an identified knowledge gap, contributes to the field, and provides novel information. Including a clear statement on the significance of the study will enhance the overall impact of the manuscript.

Organization and Clarity:
The manuscript is generally well-organized and follows a logical structure. However, I recommend that the authors consider providing a more detailed introduction and background section to provide better context for readers who may not be familiar with the species. Additionally, ensuring that the manuscript adheres to PeerJ standards and guidelines will further improve its clarity and readability.

Additional comments

In conclusion, the manuscript "Biology, Ecology, and Biogeography of the Enigmatic Desert Praying Mantis Blepharopsis mendica" presents valuable information on the species. With some minor revisions, particularly in providing additional results in the modeling section, addressing data deposition, improving language and grammar, and highlighting the significance of the study, the manuscript will be further strengthened. Overall, this research contributes to the understanding of B. mendica and its ecological characteristics.

·

Basic reporting

The manuscript is written in clear and professional English. The introduction is detailed and the literature cited is relevant. The authors cover all the papers, except the recent report of the species from the Republic of Uzbekistan:

Akhmedov A.G., Kholmatov B.R. (2019) Fauna and some ecological aspects of praying mantis (Insecta, Mantodea) of Uzbekistan. Tyumen State University Herald. Natural Resource Use and Ecology, 5(1), 129-140. DOI: 10.21684/2411-7927-2019-5-1-129-140

The structure of the manuscript conforms to PeerJ standards. Figures are of high quality and well described. All appropriate raw data are supplied by the authors, with a single exception.

The vouchers, although undoubtedly referring to the exact same specimens, are written differently on Figures and in supplementary information. They should be exactly the same and correspond to the actual vouchers pinned underneath the specimens. In addition, the COI barcode for "Pak1"/"MAN-00032", although accessible in BOLD, is absent in the supplementary Fasta file.

Experimental design

The authors convincingly demostrate how the current knowledge on Blepharopsis mendica is fragmentary, despite existing past studies dedicated to this species. Given its wide distribution and recognition in circles well beyond the specialists on Mantodea, it makes the gaps all the more glaring. The authors ask three questions: what are the biological traits of Blepharopsis mendica, what are its ecological traits and why is it distributed the way it is. To answer these, they collected field observations, lab data on the reared population, performed statistical analysis on those, obtained COI sequences, inferred the haplotype network, performed Bayesian phylogenetic analysis, molecular dating and biogeographical analysis with two models for range expansion, S-DIVA and S-DEC. The authors report most of the details of their protocols, which, together with all (see above) the original data made freely available, makes all the analyses replicable.

The methods used are well established in the field. There are some aspects of the methods that the authors should clarify directly in the manuscript. I highlighted these in the reviewed version of the manuscript as comments. Most importantly, the principle behind the delineation of the geographical regions for the biogeographical analysis should be more clear. For example, according to Figs. 6-7 there is no discontinuity in the current distribution from Morocco to Tunisia, yet the authors input these as different regions.

Validity of the findings

The results are well described and mostly supported by the data.
Regarding biological observations, there are several important details I would like the authors to be clear about, highlighted in the reviewed version of the manuscript as comments.

The ecological niche model and its discussion are sound. Although the authors were unaware of the Uzbekistan record, it neverheless falls within the boundaries of the suitable region predicted by their model. In my opinion, the authors can take it as a wonderful verification and definitely mention it in the manuscript, seeing as this population is remote from all the other data points.

However, the discussion of the results of the biogeographical analysis requires more details - as well as more caution. First of all, the sensibility of the results to the region delineation scheme is completely unexplored. The basal phylogenetic position of the sole Pakistani sample is likely to disproportionally influence the biogeographical analysis' result. The authors should discuss the possible implications if there is actually a continuum - distributional as well as genetic - in the undersampled region between the Pakistani locality and the Iranian localities. And vice versa, if the gap is real, the authors should discuss the possible historical cause. Right now the relevant discussion is way too general.
The "out of Iran" hypothesis implies that the Middle East was colonized by the species twice: first, when it expanded its distribution to Maghreb, and, second, during the latest dispersion from Iran, between which there should be an extinction. Yet a clear discussion of such implication (or lack of thereof) is absent. There are other comments highlighted in the reviewed version of the manuscript.

Otherwise, the conclusions of the authors are well stated, linked to original research question & limited to supporting results.

Additional comments

The present manuscript reports a wealth of information on the little-known desert mantid species B. mendica, including its life cycle, ootheca (egg case), defense behavior, and preferred habitat. In addition, the authors chart its haplotype network, model ecological niche and present a scenario of its historical biogeography based on a dated Bayesian phylogeny. Such comprehensive species-focused works are rare in the field, especially regarding small insect orders.

I recommend the manuscript for publication in PeerJ following minor revision.

·

Basic reporting

The literature search on the species' distribution was not exhaustive enough, leading to some flawed hypotheses in the discussion. I suggest to screan the works of Chopard, Uvarov, La Greca, Roy and others for collection data for the species.

The Figures are ok, but sometimes too small or to sharply cropped, leaving no room for the object to breathe. I suggest to re-crop the photos of the species (in particular those on deimatic display), and to arrange some of them (e.g. habitat photos) into more than one plate and make them larger. Suitable photos showing the remarkable camouflage of the species are missing.

Aside from this, the paper is well written and structured.

Experimental design

The methodology is acceptable, with the exception of a lack of proper literature study. This also concerns other taxa mentioned in the paper.

One flaw with the phylogenetic data is the overrepresentation of Iranian data, the underrepresentation of the other used regions, and the lack altogether of data from important areas like Egypt, Somalia, Sudan and Mauretania. While a literature search on these areas is not the main difficulty, obtaining genetic samples from these regions might prove difficult. Nevertheless, I suggest to at least include Egyptian samples from different areas of the country and re-do the analysis. An increase of samples from the other areas except Iran would also be welcome.

Validity of the findings

The results and their discussion mostly reflect the methodology used. The flaws in the discussion stem from the problems outlined above and from a certaiin misunderstanding of the species studied. Aridity is often suggested as drivers of vicariance in this species, while it is superbly adapted to aridity and should have benefitted from aridification. I suggest testing the hypothesis that interchanging periods of humidity (habitat regression) and aridity (habitat increase) have created the observed haplotype distribution, and enabled the species to colonize remote oasis in the Sahara and areas as far as the Thar desert.

The development of the species is only compared with Mantidae, Miomantidae and Hymenopodidae, while no member of Empusidae is mentioned, despite data being available at least for Idolomantis, Gongylus, and Empusa.

The unusual startling display reported for one female is only briefly mentioned and never revisited. I suggest to discuss that particular case in light of findings from other Mantodea.

Additional comments

The authors present a very valuable contribution on the charismatic mantid Blepharopsis mendica, which advances our knowledge on certain subpopulations of this vastly distributed Eremic species.
Detailed comments on the critical topics are found in the attached file.

Despite the issues pointed out above, I recommend this paper for publication after a major revision.

·

Basic reporting

Peer review report on manuscript Ref: Submission ID peerj- Manuscript
91411v1
Biology, ecology, and biogeography of the enigmatic desert praying mantis Blepharopsis mendica (Insecta: Mantodea)
Original submission
Recommendation: major revision

Overview and general recommendations
It is a good study. But there are some points need to correct in review file

Experimental design

Original primary research within Aims and Scope of the journal.

Validity of the findings

Impact and novelty not assessed. Meaningful replication encouraged where rationale & benefit to literature is clearly stated.

---

## Round 0.2 · accepted · Accept

The manuscript provides valuable insights into the life cycle, behaviour, habitat, and distribution of Blepharopsis mendica. The authors have conducted field observations, gathered distribution data, and performed phylogeographic analyses to support their findings. They have addressed all reviewer’s comments, and the manuscript is now well-written and ready to be published.
Congratulations.